# Research on the Road Performance of Asphalt Mixtures Based on Infrared Thermography

**DOI:** 10.3390/ma15124309

**Published:** 2022-06-17

**Authors:** Wei Chen, Kesen Wei, Jincheng Wei, Wenyang Han, Xiaomeng Zhang, Guiling Hu, Shuaishuai Wei, Lei Niu, Kai Chen, Zhi Fu, Xizhong Xu, Baogui Xu, Ting Cui

**Affiliations:** 1School of Transportation Engineering, Shandong Jianzhu University, Jinan 250101, China; chenwei13572468@163.com (W.C.); huguilingtech@foxmail.com (G.H.); xubaogui0805@163.com (B.X.); 2Zibo Transportation Management Center, Shandong Hi-Speed Company Limited, Zibo 255000, China; m17865438990@163.com; 3Science and Technology Innovation Center, Shandong Transportation Institute, Jinan 250102, China; hanwenyang@sdjtky.cn (W.H.); zhangxiaomeng@sdjtky.cn (X.Z.); shuaishuai_wei@163.com (S.W.); nlago@foxmail.com (L.N.); cck133@163.com (K.C.); fufranklin@163.com (Z.F.); xxz137152@163.com (X.X.); tingcui1016@126.com (T.C.)

**Keywords:** infrared thermal imaging technology of an unmanned aerial vehicle (UAV), optimal temperature measurement height, melt temperature threshold, molding temperature-road performance prediction model

## Abstract

Temperature segregation during the paving of asphalt pavements is one of the causes of asphalt pavement distress. Therefore, controlling the paving temperature is crucial in the construction of asphalt pavements. To quickly evaluate the road performance of asphalt mixtures during paving, in this work, we used unmanned aerial vehicle infrared thermal imaging technology to monitor the construction work. By analyzing the temperature distribution at the paving site, and conducting laboratory tests, the relationship between the melt temperature, high-temperature stability, and water stability of the asphalt mix was assessed. The results showed that the optimal temperature measurement height for an unmanned aerial vehicle (UAV) with an infrared thermal imager was 7–8 m. By coring the representative temperature points on the construction site and then conducting a Hamburg wheel tracking (HWT) test, the test results were verified through the laboratory test results in order to establish a prediction model for the melt temperature and high-temperature stability of y = 10.73e^0.03x^ + 1415.78, where the predictive model for the melt temperature and water was y = −19.18e^−0.02x^ + 98.03. The results showed that using laboratory tests combined with UAV infrared thermography could quickly and accurately predict the road performance of asphalt mixtures during paving. We hope that more extensive evaluations of the roadworthiness of asphalt mixtures using paving temperatures will provide reference recommendations in the future.

## 1. Introduction

With China’s rapid economic development and growing public demand for quality travel, the country’s total road length reached 5,198,100 km by the end of 2020 [1]. Due to the rapid development of China’s roads, the volume of traffic that must be carried on these roads has also increased, leading to a significant increase in the occurrence of damage to the road surface, resulting in travel inconveniences. Therefore, construction quality control has become a significant issue for the durability of asphalt pavements [2]. One reason for the various issues with asphalt pavements is the difficulty in controlling the temperature of the asphalt mixture during the paving process [3], as the temperature of the asphalt mixture must be strictly controlled during transportation and paving to prevent temperature segregation in the asphalt pavement. Otherwise, the road performance of the asphalt pavement will be seriously affected [4]. Thus, an increasing number of researchers have assessed the effects of paving temperatures on asphalt pavements.

In 1929, Czerny et al. [5] developed the first generation of thermal imaging cameras, which introduced the idea of active infrared thermal imaging for non-destructive testing, whereas Mahoney et al. [6,7] investigated the effects of various factors, such as the temperature difference, the density of the compacted material, transport distance, paving equipment, weather, and other variables, on the results of infrared thermography. In addition, Han, Dongdong et al. [8] used infrared thermography to monitor the heating process at the construction site of hot in-place recycling asphalt pavement to obtain the optimal heating method and temperature threshold. Hiasa et al. [9] proposed a method of data threshold analysis processing using infrared thermography, which was capable of visually measuring the delaminated areas of concrete slabs. Ellenberg et al. [10] developed a new image processing algorithm to obtain spectral images using small infrared reflections to help identify delamination, enabling the rapid identification of bridge deck delamination locations and the detection of bridge deck distress. In their study of two asphalt pavements with different aging conditions, Nishar et al. [11] used infrared spectral region data to assess defects in asphalt pavements to distinguish newly paved asphalt pavements from old asphalt pavements. Using infrared thermography on paving sites, researchers such as Nevalainen [12] found that the long transport of asphalt mixes was the leading cause of temperature segregation. Rahman et al. [13] used thermal imaging to record the temperature changes at a construction site and suggested that a temperature difference of 14 °C in the mix would result in an abnormal void fraction in asphalt pavements. Javilla et al. [14] used wheel tracking tests on AC-13 and AC-20 asphalt mixtures and concluded that the A/C index could provide a suitable indirect assessment of rutting index performance. Furthermore, Polaczyk et al. [15] tested the locking point concept by using applied performance tests and evaluating the impact of aggregate interlocking on rutting and fatigue cracking to obtain the best-defined locking point.

In this work, we used an unmanned aerial vehicle (UAV) infrared thermal imaging technique to inspect an asphalt pavement mixture paving site, to determine the optimal inspection height and to predict the performance of the asphalt pavement after grinding. In addition, to establish a new evaluation system for asphalt pavements based on infrared thermography, we provide a theoretical basis for future applications in practical engineering.

## 2. Test Methods and Principles

### 2.1. Infrared Thermal Imaging Technology of the UAV and Principles

In this study we used an M300 RTK UAV from DJI (Shenzhen, China), which was paired with a Zenmuse H20T infrared thermal imaging camera, as shown in Figure 1. Using the M300 RTK’s flight control system, the UAV could view the temperature distribution of the photographic interface on any trajectory and at any angle to obtain the required infrared images. Then, the infrared image data processing software was used to visualize the temperature distribution of the measured area. The principle of operation of infrared thermography involves the use of infrared detection elements. Thus, the infrared radiation emitted by the object was received during the test, the power signal radiated by the object was converted into an electrical signal and amplified, and then, after digital processing and image processing, a thermal image with better temperature values was finally presented on a display device [16], as shown in Figure 2.

### 2.2. Rutting Test

According to the Chinese standard JTG E20-2011 (T0719) [17], a rutting test can be used to evaluate the rutting resistance of asphalt mixtures, and its dynamic stability can more accurately reflect the ability of asphalt mixture pavements to reduce rut formation under high-temperature conditions [18]. The asphalt mixes at different melt temperatures were placed in standard test molds of 300 mm in length, 300 mm in width, and 50 mm in height, and were set up with three parallel test pieces for each compaction temperature. During the test, the specimen, along with the test mold, was placed in a wheel rutting test machine at a temperature of 60 °C ± 1 °C for no less than 5 h and no more than 12 h. The test wheel was pressed against the rutting plate specimen at 0.7 MPa, and the test wheel was rolled back and forth at a rate of 42 times ± 1 times per min for 1 h or stopped when maximum deformation reached 25 mm. Deformations d1 and d2 were recorded at 45 and 60 min.

### 2.3. Freeze-Thaw Splitting Test

Freeze-thaw splitting tests were used to analyze the water stability of asphalt mixtures at different melt temperatures, according to the Chinese Standard JTG E20-2011 (T0719) [17], by measuring the splitting tensile strength ratio of asphalt mixture specimens subjected to water damage before and after different melt temperatures and then evaluating the effect of melt temperature on the water stability of the asphalt mixtures [19]. For each asphalt mixture with different melt temperatures, eight standard Marshall test pieces were created, which were divided into two random groups of four pieces. The first group was maintained at room temperature as the control test pieces, and another group of test pieces was vacuumed for 15 min, restored to normal pressure, and placed in water for 0.5 h. Then, the test pieces were removed and placed in a plastic bag with 10 mL of water and placed in a refrigerator at a constant temperature of −18 °C ± 2 °C for 16 h ± 1 h of freezing. After the freezing was complete, the test pieces were immediately placed into a water bath at 60 °C ± 0.5 °C for 24 h after removing the plastic bag. Then, the two groups of test pieces were placed together into a constant temperature bath at 25 °C ± 0.5 °C for 2 h, for the final freeze-thaw splitting test. The splitting tensile strength ratio was calculated as follows:(1)$$R_{T1}=\frac{2p_{T1}}{\pi D_{1}h_{1}}$$
(2)$$R_{T2}=\frac{2p_{T2}}{\pi D_{2}h_{2}}$$
(3)$$TSR=\frac{{\bar{R}}_{T_{2}}}{{\bar{R}}_{T_{1}}}$$
where *R_T_*_1_ indicates group 1 splitting tensile strength specimens (MPa), *R_T_*_2_ denotes group 2 splitting tensile strength specimens (MPa), *P_T_* is the value of the test load for a single specimen (N), *D* is the Marshall specimen diameter (mm), *h* is the Marshall specimen height (mm), *TSR* is the splitting tensile strength ratio (%), ${\bar{R}}_{T1}$ is the average splitting tensile strength of group 1 specimens (MPa), and ${\bar{R}}_{T2}$ is the average splitting tensile strength of group 2 specimens (MPa).

### 2.4. Hamburg Wheel Tracking Test

The Hamburg wheel tracking (HWT) was used to evaluate the water sensitivity and resistance to rutting of the asphalt mixtures [20] according to the Chinese standard JTG E20-2011 (T0719) [17]. The asphalt pavement was core-cut on site, and the core samples were 150 mm in diameter and 38 mm thick. Then, standard specimens were cut and placed in the Hamburg test mold. When the water tank temperature reached 50 °C, the steel wheel with a wheel load of 705 N ± 4.5 N was reciprocally crushed at a speed of 52 ± 2 times per minute. The maximum speed of the wheel through the midpoint of the specimen was 0.305 m/s. When the loading time reached 20,000, the test steel wheel was automatically lifted, at which point the rutting depth was recorded.

## 3. Field Measurement Basis with UAV Infrared Thermography

### 3.1. Project Overview

This work was based on the reconstruction and expansion project of a section of a highway in Shandong Province, China, which was expanded from four to eight lanes in both directions, with a design speed of 120 km/h. This study combined the actual asphalt pavement top layer paving site conditions with UAV infrared thermography to measure the temperature of the asphalt pavement during paving, to predict the actual road performance of the asphalt pavement after grinding was complete.

### 3.2. Identifying the Optimal Temperature Measurement Height

The Zenmuse H20T imaging system was affected by the ambient temperature, humidity, object emissivity, and the measurement height. Therefore, to investigate the accuracy of temperature measurement results at different heights, it was necessary to carry out temperature measurement studies at different heights on the asphalt pavements in the same area, to determine the optimal height for temperature measurements.

#### 3.2.1. Parameter Calibration

Before testing, it was necessary to set the instrument parameters according to the site environment, including the ambient temperature, ambient humidity, emissivity, and measurement height [21]. The temperature at the site was 17 °C, the relative humidity was 58%, and the emissivity of each object is shown in Table 1. The parameters were imported into DJI Thermal Analysis Tool software, where the images showed the temperature data for the desired location.

#### 3.2.2. Temperature Measurement Results and Analysis of Different Measuring Heights

To determine the optimal temperature measurement height for the UAV with the infrared thermal imager, we randomly selected an area where grinding was complete and which retained the residual temperature. We placed four objects with a lower temperature in this area to form a 1 m × 1 m detection area and obtained an infrared thermal image of the area using the UAV thermal imager, as shown in Figure 3. Starting at a distance of 3 m from the pavement, the measurement height was continuously adjusted upwards at 1 m intervals to investigate the pattern of temperature change with the shooting height, to determine the optimal height for temperature measurements. To prevent heat loss caused by external environmental factors during the temperature measurement process, the temperature acquisition time was limited to 20 s to ensure detection accuracy as much as possible.

According to the initial on-site measurements, when the temperature measurement height was more than 12 m, the temperature measured by the UAV infrared thermal imager was more dissimilar than the temperature measured by the handheld temperature measurement gun, which deviated from the actual temperature of the asphalt pavement; thus, a temperature measurement height range of 3–11 m was selected. By entering the actual height data for the temperature measurements in the DJI Thermal Analysis Tool software, the actual temperature of the measured area could be displayed directly, as shown in Figure 4. We chose six temperature points from each measurement area and obtained the average value as a representative of the temperature at this height.

For each height, the maximum, minimum, and average measured temperatures were analyzed to investigate their variation patterns and to determine the optimal height for temperature measurements, as shown in Figure 5, Figure 6 and Figure 7.

As shown in Figure 5, Figure 6 and Figure 7, the distribution curves of the maximum and average temperatures followed the same pattern of low change. Although fluctuations in temperature occurred between the different heights, the variations in temperature between the adjacent heights were minor and did not exceed 1 °C. The maximum temperature fluctuated around 35 °C, whereas the average temperature was around 34 °C. The variations in the minimum temperature curve were different; however, the temperature fluctuated around 30 °C. With increasing measurement height, the measurement area increased and was more subject to various uncertainties. The combined temperature data showed that the measured temperature values were all single-point, and the maximum and minimum temperatures were not the same between the different heights in the same area. Hence, the average temperature was more representative. When the measurement height was between 7 and 8 m, the three curves were close to each other, and the variations in each temperature were the smallest. Combined with the measurement results of the point-type temperature-measuring gun, the best temperature measurement height was determined to be 7–8 m.

### 3.3. On-Site Measurements

During the construction of the top layer of the asphalt pavement, the site temperature was 20 °C and the distance from the mixing plant to the construction site was 3 km. The tipper truck delivered the asphalt mix to the paving site and then poured it into the spreading machine’s receiving hopper, where it was transported via a conveyor to the spreading machine for secondary mixing and finally paving. During this process, multiple UAVs with infrared thermal imagers measured the temperatures of the dump trucks and the spreading machine throughout the process, as shown in Figure 8. Infrared thermography was used to measure the temperature distribution of the asphalt mixture during the paving process, focusing on the pavement that had just been paved but which was not yet ground, as shown in Figure 9.

Due to the slow travel speed of the spreading machines, it was not possible to measure the temperature data over long distances at once. Therefore, the UAV thermal imager had to remain in the air for continuous temperature measurements and then the captured data images were combined, which reduced the temperature losses during the measurement process. After measurement, a randomly selected area of 50 m in length and 7.5 m in width was studied and analyzed. The research area was divided into 0.5 m × 0.5 m squares [2] with 1500 squares, with the average of the temperatures measured in each square taken as the measured value for that square, as shown in Figure 10.

As shown in Figure 10, the temperature distribution of the SMA-13 asphalt mix pavement after paving was very uneven, with a large temperature span. The Chinese standard JTG F40-2004 [22] specifies that the paving temperature has to be greater than or equal to 160 °C, and the temperature at the start of the initial grind must be greater than 150 °C. In the research area, the maximum temperature was 168 °C, and the average temperature was 155 °C, with the lowest temperatures (142 °C) located in the marginal parts. Therefore, we inferred that the proximity of this area to the shoulder resulted in edge temperatures that did not meet the specification requirements [22], and temperature segregation could occur, affecting the road performance.

According to Figure 9 and Figure 10, the reasons for the uneven paving temperatures were twofold. First, the asphalt mixture after a certain distance of transport and the surface temperature of the asphalt mixture in the delivery truck dropped considerably, whereas the internal temperature loss of the mixture was small; thus, the temperature difference of the asphalt mixture was significant. Second, the asphalt mixture dumped from the delivery truck entered the spreading machine receiving hopper. The spiral distributor evenly paved it, and this process is exposed to air; thus, temperature dissipation inevitably occurred.

To reduce the significant temperature differences that occurred in the asphalt mixes during transport, reducing the heat dissipation from the asphalt mixes was necessary. During transport, temperature segregation was more severe at the top of the truck and on both sides of the carriage; thus, attention had to be paid to strengthening the insulation of the internal structure of the material carrier. To avoid temperature segregation, when loading the material carriage, the following method was used: the front part was loaded first, then the rear part, and finally the middle part. The bottom of the carriage had to be coated with a lubricant to achieve unloading of the entire asphalt mixture. During the paving process, spacing between the spiral blades on both sides could be reduced to an unequal distance so that the asphalt mixtures on both sides could be mixed evenly through the spiral blades, reducing the temperature difference. In addition, it was possible to adjust the size and tilt angle of the spiral blades to avoid heat exchange between the spiral blades and the atmosphere, as well as to reduce the temperature dissipation of the asphalt mix. It was also possible to adjust the height of the spiral distributor to speed up the flow of the asphalt mixture so that low-temperature aggregates could be quickly remixed with high-temperature aggregates, thus reducing the occurrence of uneven temperature distributions in the asphalt mix pavements [23].

### 3.4. Evaluation of Road Performance of Asphalt Mixtures Based on the HWT Test

To investigate the effects of different paving temperatures on the road performance of the SMA-13 asphalt pavements, the asphalt pavements were marked at different paving temperatures, and after grinding, the marked locations were cored according to the HWT test to analyze their road performance. According to Figure 10, the research area was divided into five temperature gradients: 130–140 °C, 140–150 °C, 150–160 °C, 160–170 °C, and 170–180 °C. Values of 135 °C, 145 °C, 155 °C, 165 °C, and 175 °C were selected as representative values for each temperature gradient, and then the cores were taken, cut, and subjected to HWT testing. Because the area selected was close to the road border, the overall temperature was slightly lower, with few areas where the paving temperature reached 175 °C. Therefore, only the core samples from areas at 135 °C, 145 °C, 155 °C, and 165 °C were selected for the Hamburg wheel mill test, and the test results are shown in Figure 11.

As shown in Figure 11, the rutting depths at 10,000 and 20,000 wheel rolls met the requirements of the Chinese specifications of ≤4 mm and ≤10 mm [17], and the paving temperatures of 135 °C, 145 °C, 155 °C, and 165 °C at a wheeling depth of 20,000 were 3.576, 3.472, 2.639, and 2.401 mm, respectively. The rutting depth at a paving temperature of 135 °C increased by 48.9% compared to that at 165 °C. The rutting depth at the 145 °C positions increased by 35.5% compared to that at the 165 °C positions, whereas the rutting depth at the 155 °C positions increased by only 3.0% compared to that at the 165 °C positions. This indicated that when the paving temperature of the asphalt pavement was below 155 °C, temperature segregation may have occurred, and its resistance to rutting was significantly reduced after grinding, affecting the performance of the road.

The analysis showed that the void ratio was large at the location where temperature segregation occurred. A good nested structure could not form between the aggregates, resulting in a greater-than-standard rutting depth under the action of wheel rolling, which in the long run would form ruts and affect the service life of the pavement [18]. Therefore, for the temperature segregation produced in the construction of the asphalt pavement, although the final quality acceptance of the pavement performance indicators could meet the requirements, to increase the service life of the pavement, the construction process could be used to monitor the paving temperature of the asphalt mixture using UAV infrared thermal imaging technology. To extend the service life of the road, for areas below 155 °C, there should be a timely adjustment of the paver parameters to reduce the occurrence of the temperature segregation phenomenon.

## 4. Effect of Melt Temperature on the Road Performance of Asphalt Mixtures

The range of the temperature distribution during asphalt paving was obtained via infrared thermographic inspection of the SMA-13 asphalt mixes at the construction site. Due to the construction site conditions, it was impossible to cut many rutting plate specimens; thus, it was difficult to accurately evaluate the rutting resistance, water stability, and other road properties of the asphalt mixtures at different paving temperatures. In this study, the paving temperature of the asphalt mixture in the field was combined with the road performance measured in the indoor tests, with the paving temperature at the construction site set as the mixing and melt temperature for the indoor asphalt mixture tests. The prediction model of asphalt mixture road performance and melt temperature was established through indoor tests to improve the accuracy and efficiency of road performance testing.

### 4.1. Indoor Experimental Design

In this study, the asphalt mixes were mixed at 135 °C, 145 °C, 155 °C, 165 °C, and 175 °C to simulate the paving temperatures in the field and analyze the high-temperature stability and water stability of the mixes at these temperatures. The results of the tests were compared with those of the HWT test, to verify the effect of SMA-13 asphalt mix paving temperature on the on-road performance.

#### 4.1.1. Raw Material

The use of aggregates and asphalt brought back from the construction site for the indoor tests prevented errors from arising due to differences in the test materials. The performance specifications of the materials used in this work are presented below.

(1)
*Asphalt*


The SBS modified asphalt was selected to meet the requirements of the Chinese Code [15], and the test results are shown in Table 2.

(2)
*Coarse aggregate*


The coarse aggregate consisted of lava rock, with 5–10 mm and 10–15 mm sizes. The physical properties of the coarse aggregate were determined in accordance with the Chinese specification JTG E42-2005 [24], and the technical indicators satisfied the specification, as shown in Table 3.

(3)
*Fine aggregates*


The fine aggregates in the asphalt mix consisted of 0–3 mm limestone, as shown in Table 4.

#### 4.1.2. Gradation Composition

According to the technical indicators of the production mix ratio of the asphalt mix for the top layer of the expansion project, it was known that the optimal asphalt content was 5.9% and the amount of fiber was 0.35%. The composition of the grade is shown in Table 5.

### 4.2. Effect of Melt Temperature on the High-Temperature Stability of Asphalt Mixtures

As shown in Figure 12, the wheel tracking test was used to evaluate the high-temperature stability of the asphalt mixture specimens at different melt temperatures. By measuring the dynamic stability and rutting depth of the asphalt mixture specimens at different melt temperatures, the effect of melt temperature on the rutting resistance of the asphalt mixture could be evaluated. The test results are shown in Figure 13.

The dynamic stability of the asphalt mixture specimens at different melt temperatures in the wheel tracking test satisfied the requirements of JTG E20-2011 (T0719), as shown in Figure 13 [17]. The rutting depth decreased as the melt temperature of the specimen increased, with the rutting depth decreasing by 0.15 mm as the melt temperature increased from 135 °C to 155 °C. The rutting depth was reduced by 0.73 mm when the melt temperature increased from 155 °C to 175 °C, and as the melt temperature gradually increased, the dynamic stability showed a steady increase, reaching 4221 cycles/mm when the melt temperature was 175 °C.

In this case, for the asphalt used in the SBS modified asphalt, the modifier inside the asphalt was cured and did not flow as well at lower temperatures, which caused the inter-aggregates not to be filled with bitumen and to exhibit a large internal void ratio, resulting in poor high-temperature stability. When the melt temperature increased, the asphalt fully wrapped the aggregate, bonding between the aggregates was tight, and compaction gradually increased after compaction, which resulted in an asphalt mixture with increased rutting resistance. Therefore, the more significant the difference in the melt temperature, the larger the difference in the internal voids of the mix, and the more likely it was that pressure-tight rutting would occur with traffic.

### 4.3. Effect of Melt Temperature on the Water Stability of Asphalt Mixtures

The indoor tests utilized freeze-thaw splitting tests to analyze the water stability of the asphalt mixes at different melt temperatures. This was carried out by calculating the freeze-thaw splitting strength ratios before and after water damage to the five groups of asphalt mix specimens with different melt temperatures and then evaluating the effect of the melt temperature on the water stability of the asphalt mixes. The specific test results are shown in Figure 14.

As shown in Figure 14, when the melt temperature was 135 °C, the freeze-thaw splitting strength ratio was 68.8%, which did not meet the specification requirements. However, when the melt temperature was above 145 °C, the freeze-thaw splitting strength ratios of all five groups of specimens met the specification requirements. With an increase in the melt temperature, the freeze-thaw splitting strength ratio of the asphalt mix increased slightly; however, the increase was minimal. When the melt temperature was 175 °C, the freeze-thaw splitting strength ratio of the asphalt mix had a maximum value of 88.9%. This showed that the effect of melt temperature on the freeze-thaw splitting strength ratio was small, and the asphalt mixture specimens at different melt temperatures can maintain good water stability under freeze-thaw cycles, making the pavement less susceptible to water damage. This was because the melt temperature was more significant than 145 °C, the aggregate was covered by the asphalt, and the adhesion strength of the asphalt to the aggregate was greater, which improved the water stability of the asphalt mix.

### 4.4. Research on the Predictive Modeling of Melt Temperature and Road Performance

According to the indoor test results of the asphalt mixtures with different melt temperatures, the asphalt mixture road performance during the paving of the SMA-13 asphalt pavement was simulated, and the relationship between the melt temperature and the dynamic stability and freeze-thaw splitting strength ratio was established. In the construction of asphalt pavements, UAV infrared thermography was used for testing, thus predicting the road performance of the asphalt mix.

#### 4.4.1. Building the Model for Melt Temperature/High-Temperature Stability Prediction

Using UAV infrared thermal imaging technology to detect the melt temperature at the construction site, and combining the results with the dynamic stability results obtained from the indoor tests, we established a prediction model between the asphalt pavement melt temperature and dynamic stability. This enabled the dynamic stability distribution of the asphalt pavements to be visually observed in real-time during the construction process, thus allowing us to evaluate the high-temperature stability of the asphalt pavements. To establish the relationship between the melt temperature and dynamic stability, each melting temperature was divided into five intervals: 130–140 °C, 140–150 °C, 150–160 °C, 160–170 °C, and 170–180 °C. Based on the temperature distributions during the paving process, the temperature intervals were equally distributed in combination with the dynamic stability measured in the room to establish a prediction model for melt temperature/dynamic stability, as shown in Figure 15.

As shown in Figure 15, the predictive model for melting temperature/dynamic stability followed y = 10.73e^0.03x^ + 1415.78, R^2^ = 0.98; therefore, in the process of asphalt pavement construction, the melt temperature could be used to further predict the dynamic stability distribution of the asphalt pavement through temperature acquisition by means of UAV thermal imaging technology. This would make it possible to quickly determine whether the high-temperature stability of asphalt pavements would meet the specification requirements during construction, significantly improving the efficiency of testing the quality of asphalt pavement construction.

#### 4.4.2. Building the Model for Forming Temperature/Water Stability Predictions

According to the experimental design, each paving temperature was divided into five temperature intervals, and the five paving temperature intervals were equated to the melt temperature of the freeze-thaw splitting test to obtain a predictive model of the melt temperature to the freeze-thaw splitting strength ratio during construction, as shown in Figure 16. This allowed the water stability of asphalt pavements to be further evaluated based on the distribution of paving temperatures during construction.

As shown in Figure 16, the predictive model for the resulting melting temperature/freeze-thaw splitting strength ratio was y = −219.18e^−0.02x^ + 98.03. During construction, the freeze-thaw splitting strength ratio of the asphalt mixture could be quickly predicted based on the paving temperature to evaluate whether the water stability of the asphalt pavement could meet the requirements of the standard JTG E20-2011 (T0719) [17]. For locations that would not satisfy the requirements, adjustments could be made when appropriate.

### 4.5. Evaluation System of Asphalt Mixture Road Performance Based on Infrared Thermography

Using UAV infrared thermography on the paving site of asphalt pavement, the temperature distribution of freshly paved, but not yet rolled, asphalt pavement was obtained. Then, for the representative temperature points, we cored, cut, and sent the samples indoors for HWT testing to obtain the water and high-temperature stability of the asphalt mixture at each representative melt temperature point. The materials used at the construction site were used in the indoor tests to obtain the high-temperature stability and water stability of the asphalt mix at representative melt temperature values. By comparing these two datasets, we found that the high-temperature stability and water stability of the asphalt pavement met the requirements of the JTG E20-2011 (T0719) specification [17] after grinding, and during grinding, the paving temperature was more significant than at 155 °C. However, paving temperatures below 145 °C required timely adjustments to the paving process on-site to reduce the potential for later pavement distress.

The above tests were successfully carried out. Before on-site paving operations, the aggregates on-site were subjected to indoor tests in advance to establish a predictive model of the melt temperature/road performance applicable to the section. According to the minimum roadworthiness specified by the design unit, the threshold value for the melt temperature could be calculated. The melt temperature thresholds obtained from the indoor tests were converted into paving temperature thresholds during the paving process. Then, real-time temperature monitoring of the paving site was carried out using UAV infrared thermal imaging technology, with the UAV’s temperature measurement height controlled at 7–8 m. Manual intervention was carried out for areas where temperatures were lower than the threshold to reduce post-maintenance costs. For areas where temperatures were close to the threshold, stakes and locations were recorded, and future essential pavement maintenance could be carried out on the area to prevent early damage and increase the lifespan of the road. Based on this method, the paving process could be quickly tested and the road performance of the asphalt pavement could be evaluated in real-time.

## 5. Conclusions

In this work, we combined UAV infrared thermography with laboratory tests to investigate how to more efficiently and accurately analyze the high-temperature stability and water stability of asphalt pavements during the paving process, which resulted in the following conclusions.

(1)Using UAV infrared thermal imaging technology to continuously measure the temperatures at different heights in the same area, and by studying the highest temperature, lowest temperature, and average temperature of the temperature measurement area, we obtained the best temperature measurement height of 7–8 m for UAV infrared thermal imaging technology.(2)For the road section in this study, when the melt temperature increased from 135 °C to 155 °C, the rutting depth decreased by 0.15 mm and the freeze-thaw splitting strength ratio increased by 24.3%. When the melt temperature increased from 155 °C to 175 °C, the rutting depth decreased by 0.73 mm and the freeze-thaw splitting strength ratio increased by 4.1%. Therefore, infrared thermography could be used to monitor the paving temperature of asphalt pavements during paving, and quick predictions of road performance could be made.(3)The prediction model for melt temperature/dynamic stability was obtained as y = 10.73e^0.03x^ + 1415.78, where the prediction model for the melt temperature-freeze-thaw splitting strength ratio was y = −219.18e^−0.02x^ + 98.03, which could be used to predict the field paving temperature threshold.

## Figures and Tables

**Figure 1 materials-15-04309-f001:**
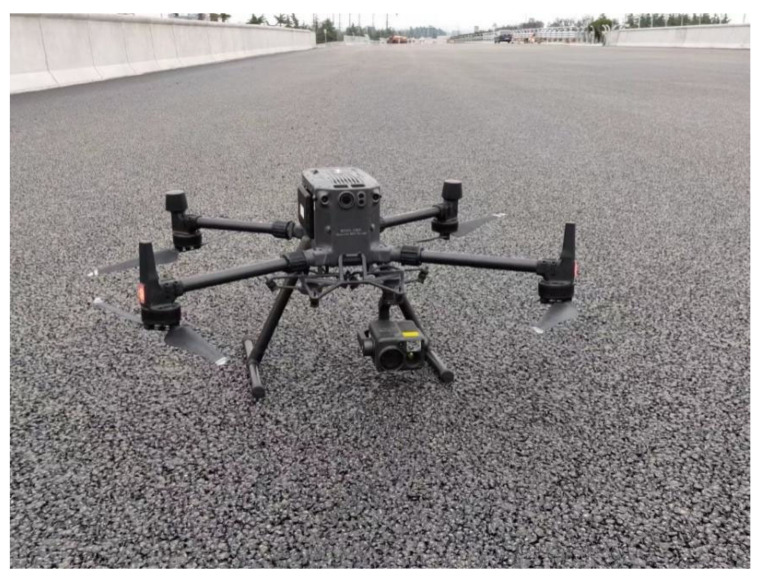
M300 RTK UAV.

**Figure 2 materials-15-04309-f002:**
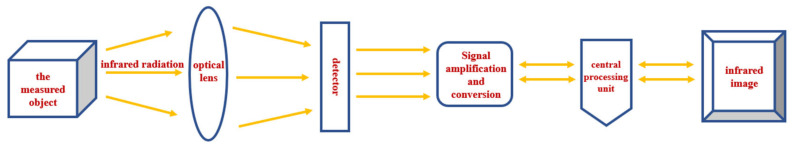
Principal infrared thermal imaging instrument temperature measurement diagram.

**Figure 3 materials-15-04309-f003:**
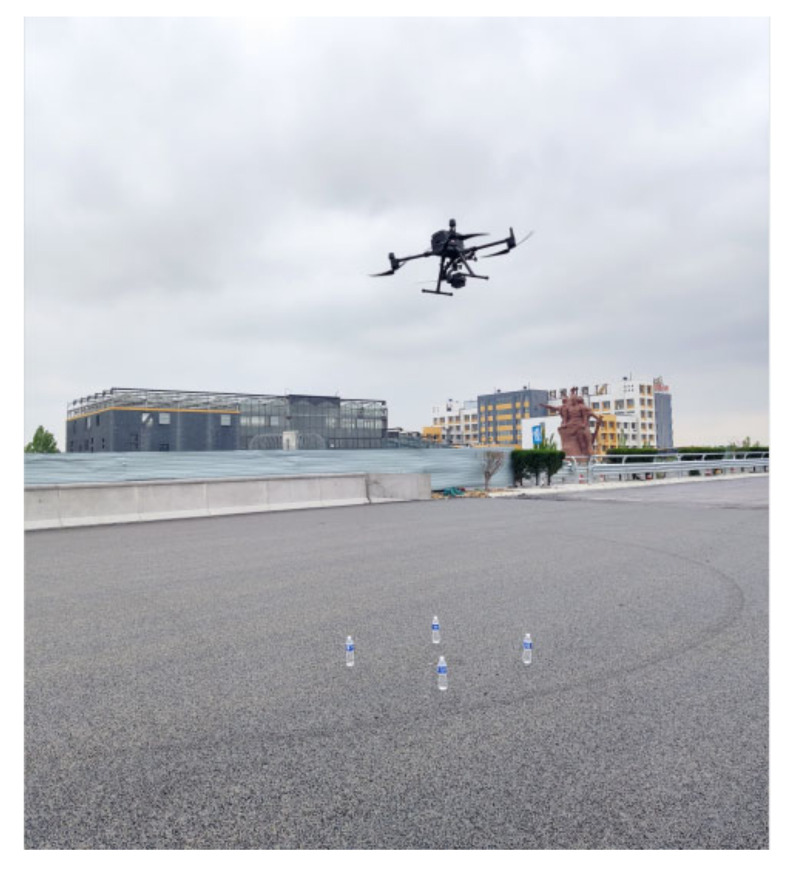
On-site temperature measurements.

**Figure 4 materials-15-04309-f004:**
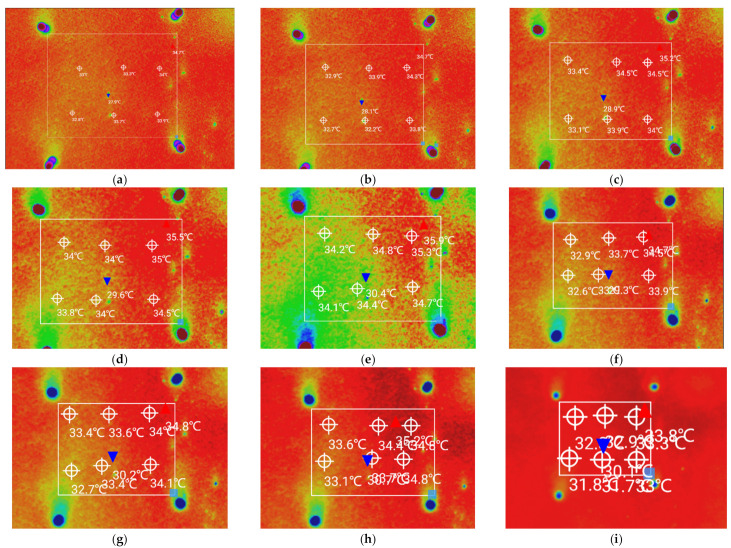
Infrared images at different measurement heights: (**a**) h = 3 m, (**b**) h = 4 m, (**c**) h = 5 m, (**d**) h = 6 m, (**e**) h = 7 m, (**f**) h = 8 m, (**g**) h = 9 m, (**h**) h = 10 m, and (**i**) h = 11 m.

**Figure 5 materials-15-04309-f005:**
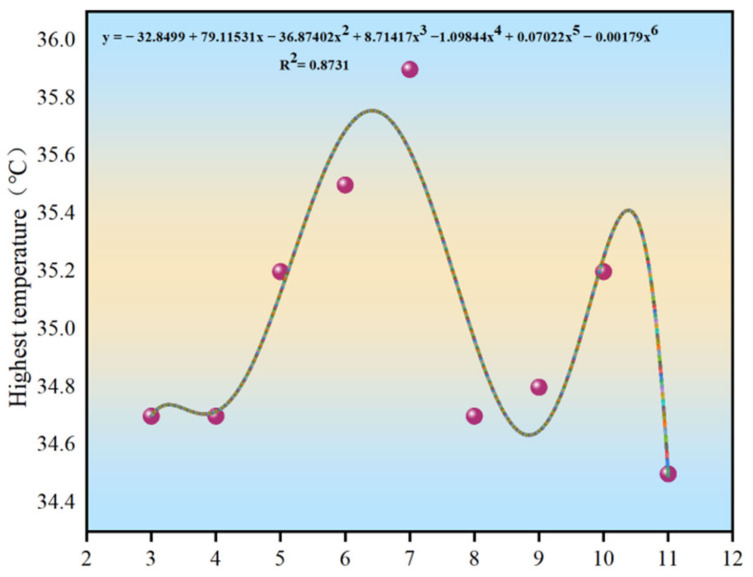
Distribution curve of the maximum temperature.

**Figure 6 materials-15-04309-f006:**
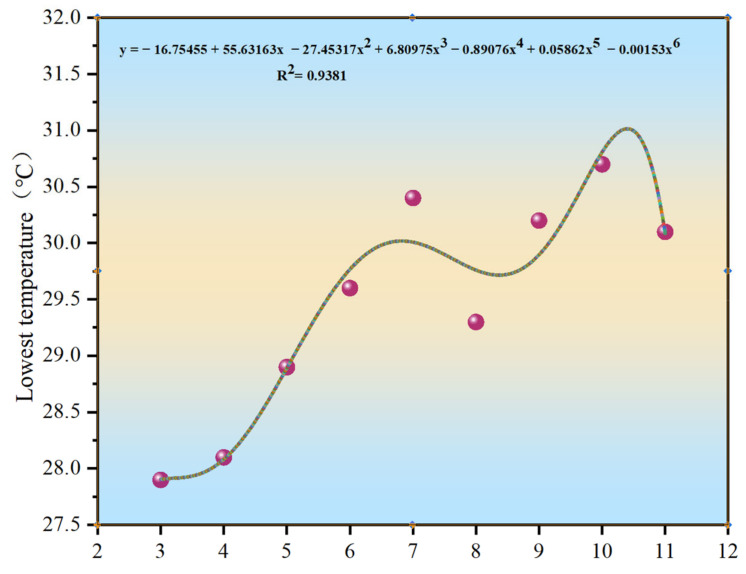
Distribution curve of the minimum temperature.

**Figure 7 materials-15-04309-f007:**
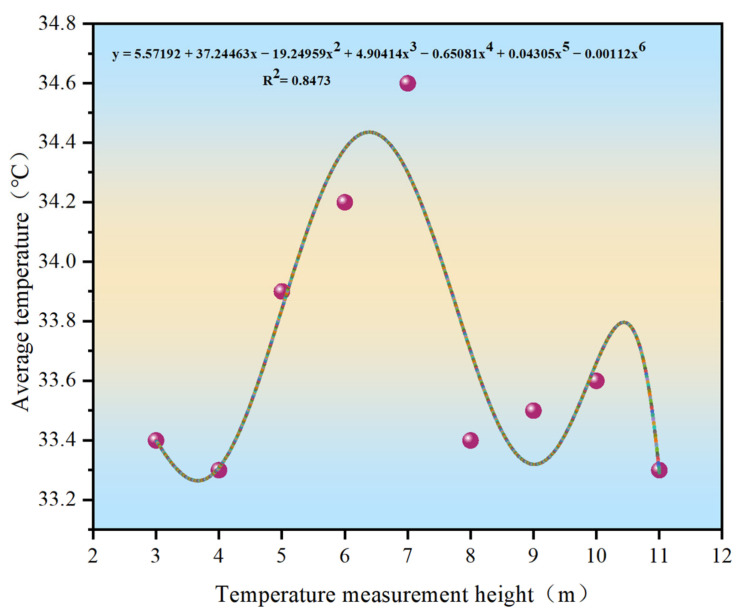
Distribution curve of the average temperature.

**Figure 8 materials-15-04309-f008:**
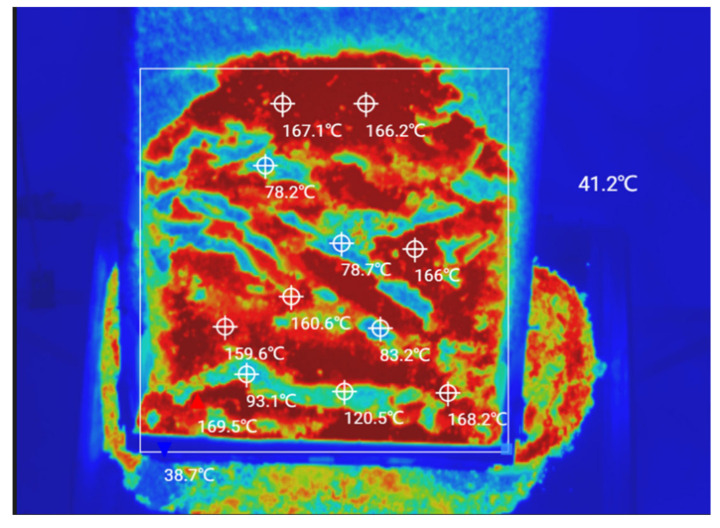
Infrared thermal image of the asphalt mix on the tipper truck.

**Figure 9 materials-15-04309-f009:**
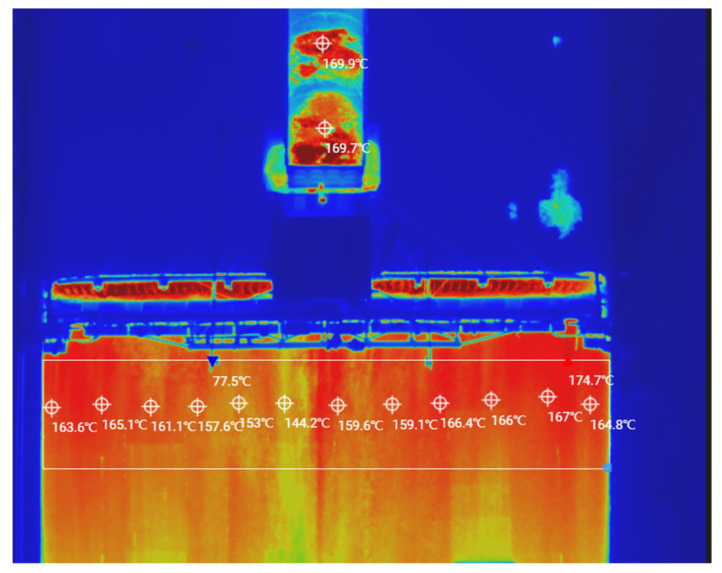
Infrared thermal image of asphalt pavement top layer construction.

**Figure 10 materials-15-04309-f010:**
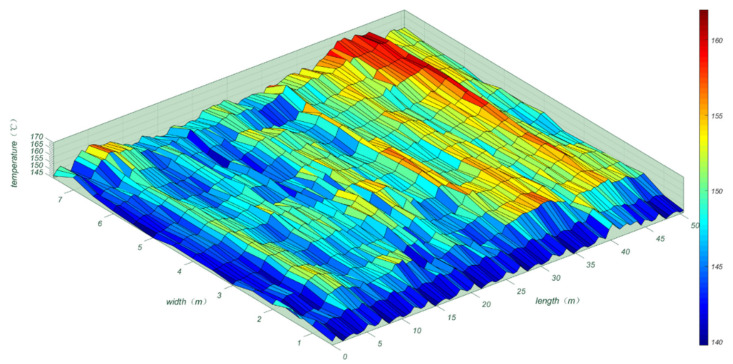
Study area temperature distribution map.

**Figure 11 materials-15-04309-f011:**
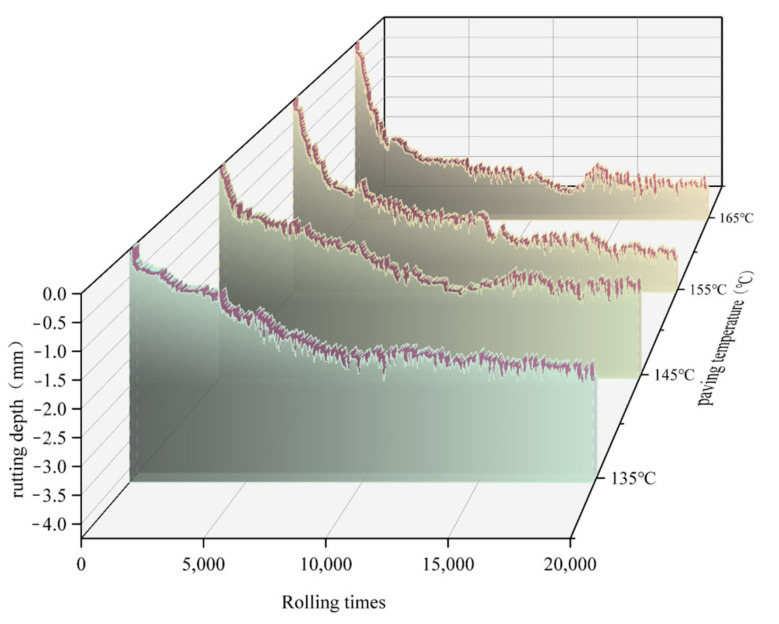
Results of the HWT test.

**Figure 12 materials-15-04309-f012:**
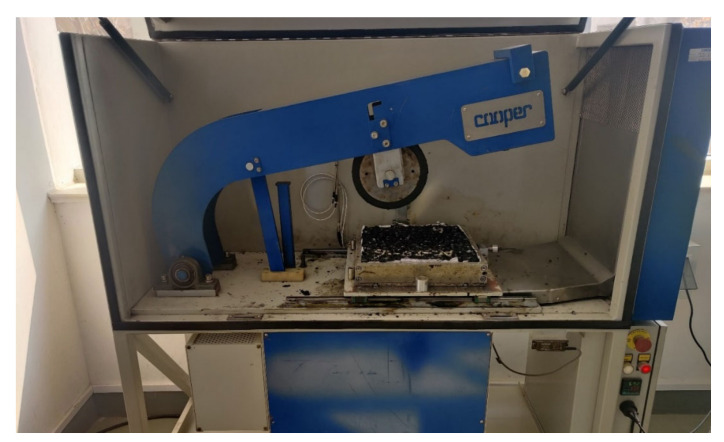
Wheel tracking test.

**Figure 13 materials-15-04309-f013:**
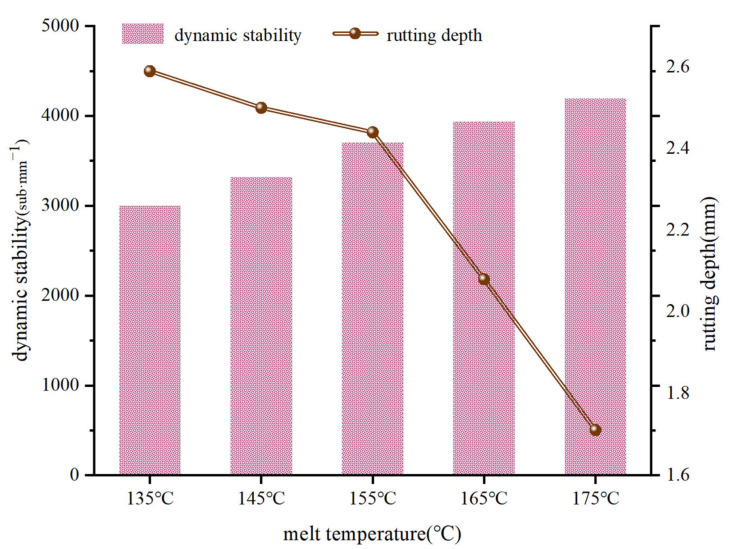
Results of the wheel tracking tests on SMA-13 asphalt mixtures with different melt temperatures.

**Figure 14 materials-15-04309-f014:**
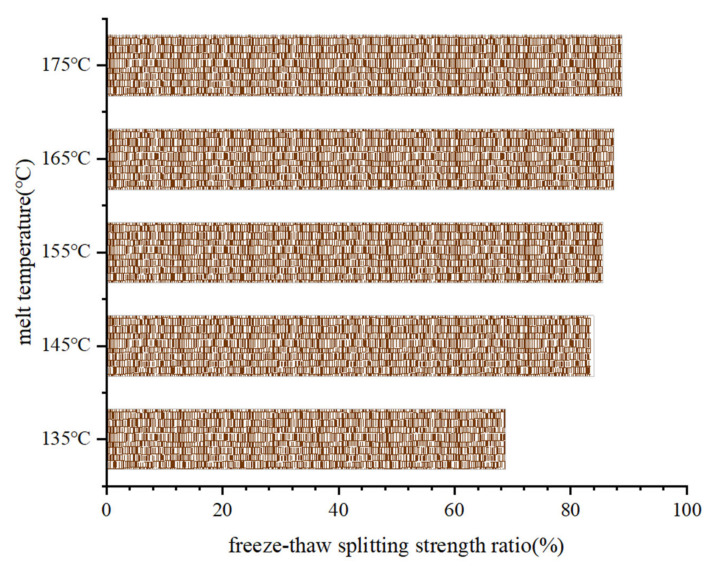
Results of the freeze-thaw splitting tests on the SMA-13 asphalt mixes with different melt temperatures.

**Figure 15 materials-15-04309-f015:**
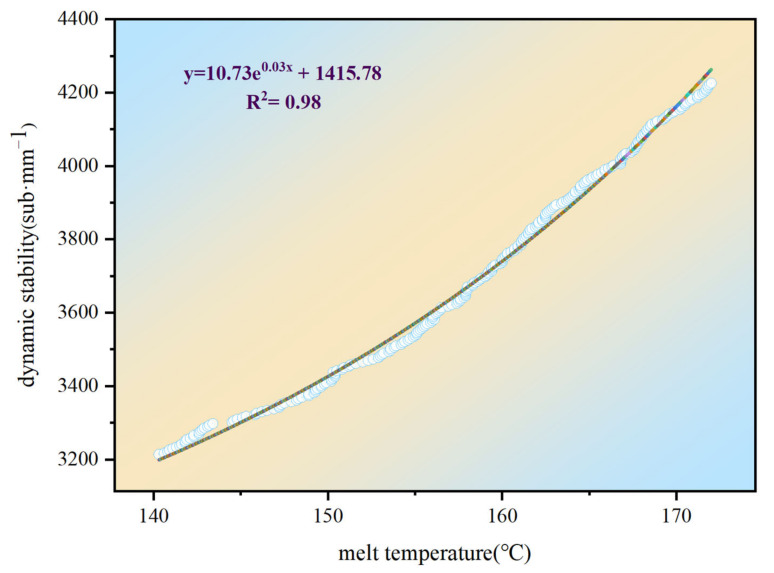
Predictive model for melt temperature/dynamic stability.

**Figure 16 materials-15-04309-f016:**
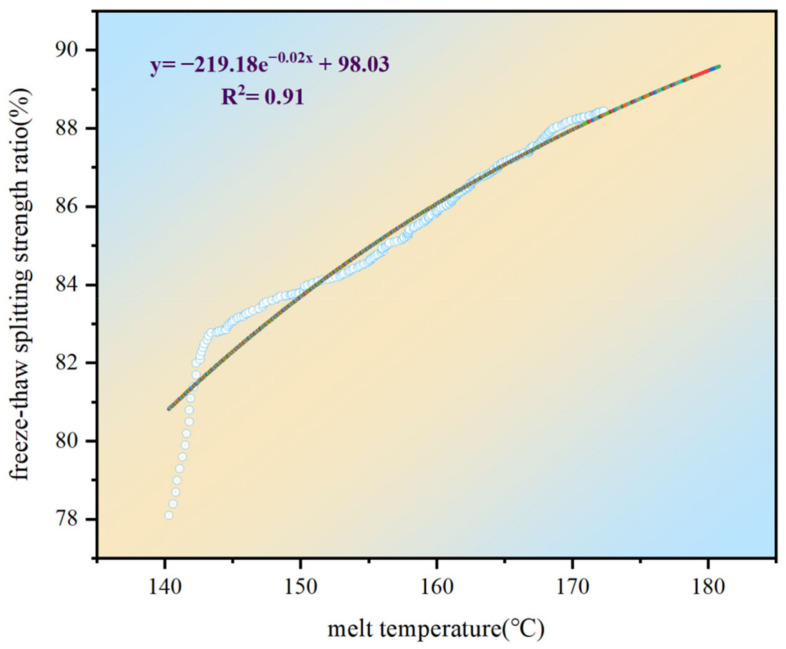
Predictive model for the melt temperature/freeze-thaw splitting strength ratio.

**Table 1 materials-15-04309-t001:** Emissivity of various substances.

Materials	Emissivity
Asphalt	0.97
Water	0.95
Iron	0.94
Cement	0.96
Basalt	0.72
Concrete	0.94

**Table 2 materials-15-04309-t002:** Technical indices of the SBS modified asphalt.

Item	Specification	Test Values
Penetration (25 °C, 0.1 mm)	40–60	53
Softening point (°C)	≥60	77.0
Ductility (5 °C, cm)	≥20	30
Dynamic viscosity at 135 °C (Pa·s)	≤3	1.3

**Table 3 materials-15-04309-t003:** Technical indices of the coarse aggregate.

Items		Specification	Test Values
Crush value (%)		≤26	9.7
Apparent relative density (g/cm^3^)	10–15 mm	≥2.6	2.926
	5–10 mm		2.967
Water absorption (%)	10–15 mm	≤2.0	1.6
	5–10 mm		1.2
Abrasion value (%)		≤28	9.5

**Table 4 materials-15-04309-t004:** Technical indices of the fine aggregates.

Items	Specification	Test Values
Apparent relative density (g/cm^3^)	≥2.5	2.955
Water absorption (%)	-	1.7
Sand equivalent (%)	≥60	62.7
Angularity (s)	≥30	39

**Table 5 materials-15-04309-t005:** SMA-13 gradation composition.

Seive size(mm)	16	13.2	9.5	4.75	2.36	1.18	0.6	0.3	0.15	0.075
Passing rate(%)	99.8	91.6	61.7	25.6	20.6	17.9	15.9	14.5	12.1	9.8

## Data Availability

Data sharing is not applicable for this article.

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
