# Peer review of "Research on the Road Performance of Asphalt Mixtures Based on Infrared Thermography"

_materials, 2022, doi:10.3390/ma15124309_

Round 1

Reviewer 1 Report

The ABSTRACT section is well–structured, is informative.

The aims and objectives of the research are well defined. The topics (quality of asphalt pavement construction, UAV infrared thermal imaging technology, paving process & road performance) are important research areas.

The TITLE is well chosen, it is comprehensive and defines the research carried out.

The manuscript is structured properly and have the basic structure of a typical research paper (INTRODUCTION, MATERIALS & METHODS, RESULTS & DISCUSSIONS, CONCLUSIONS, REFERENCES, etc.). The paper is well–structured and its parts are logically interconnected. Also, the subchapters are well–structured. Overall, this manuscript is well–written and interesting to read.

The INTRODUCTION is quite succinctly presented, but on the whole, it is comprehensive with the state of research in the field (asphalt pavement, asphalt mixture, inspection). This section provide the necessary background information needed to understand the paper (state of art, UAV infrared thermal imaging technique for inspection, asphalt mixture pavements, effect of various factors, predict the performance). The authors give informations on the studied problem, and cite other studies that have obtained relevant results. The review of the literature presented in this section is, in general, adequate.

The MATERIALS & METHODS section (TEST METHODS & PRINCIPLES) is relatively well described and include detailed information about the implementation details (i.e. Infrared Thermal Imaging Technology, UAV principles, Splitting Test, HWT Test). This section include all the technical details of the experimental setup, measurement procedure, and details of how the methods were validated. Overall, this section is technically and fairly detailed.

The RESULTS & DISCUSSIONS section is well–structured. This section describe the important RESULTS of the research, using the measurements with UAV infrared thermography, followed by several DISCUSSIONS upon the on–site measurements, Evaluation of the road performance of asphalt mixtures, indoor tests, prediction model, representative images & graphs, including the predictive modelling related aspects (on melt temperature and road performance, temperature stability prediction, water stability prediction, and evaluation system).

The CONCLUSION section succinctly summarize the major points of the paper, derived from the RESULTS and the DISCUSSIONS. The authors fairly concludes in just a few sentences given the rich discussion in the body of the paper, is quite succinctly & very briefly presented, without other opinions, conclusions or remarks. It is only technical comments. I would recommend presenting the main characteristics and the major conclusion that individualize this research.

The list of REFERENCES is long and relatively well chosen. The entire bibliography is current (the oldest being from 2008, with one exception - poz. 21 which is from 1987).

The graphic addenda is remarkable. The Tables are representative and the Figures & Graphs have good qualities.

Pay attention to grammatical mistakes. A review and correction is required in a few places. pay attention to the hyphenation (syllabification). Maybe this function should be dropped for the entire text.

Reviewer 2 Report

Dear authors,

I appreaciate your paper and I belive it can give useful information during the asphalt pavement construction. Please find below some minor comments:

line 82. What you mean with "more fantastic" 

chapter 2.3: please provide more information about the way you run the test (i.e. load rate, formula of splitting tensile strength etc.)

Figure 6: how do you explain the differnce of pattern? it seems that the minimum temperature tends to increase with height measurement

Line 273-276. Precision at thousands for the percentage is not realistic. Please put just 1 number after the dot (i.e. 48.9% insted of 48.938)

I would not talk of predictive model, but mostly of an interpolation for a specific type of asphalt (SMA-13). I think that an exhaustive model would take into account different types of asphalt. 

Reviewer 3 Report

  • You should improve the language of the manuscript.
  • The abstract should be improved. Especially the flow of the language. I should start with a short introduction, objectives, and major conclusions.
  • Your introduction needs more information about performance. I missing some general information regarding the performance of asphalt mixtures, such as https://doi.org/10.1016/j.conbuildmat.2017.07.164 and doi.org/10.1080/14680629.2021.1908408.
  • Line 35: it sounds weird, you talking about mileage and then you use km. And it is probably more than 519.81 km. Thousands?
  • Line 291: What is forming temperature?
  • Table 5: it says nothing. Include full gradation instead.
  • Please provide mix design data.
  • You don`t have real conclusions. Don`t repeat your discussion.

Round 2

Reviewer 3 Report

Thank you for addressing my comments.